# Anionic Polyelectrolyte Hydrogel as an Adjuvant for Vaccine Development

**Mariya Kozak** [1],*[ID]**, Nataliya Mitina** [2][ID]**, Alexandr Zaichenko** [2] **and Vasyl Vlizlo** [1]

[1] Institute of Animal Biology of the National Academy of Agrarian Sciences (NAAS) of Ukraine, 79034 Lviv, Ukraine; buiatricsua@gmail.com

[2] Department of Organic Chemistry, Lviv Polytechnic National University, 79013 Lviv, Ukraine; nmitina10@gmail.com (N.M.); zaichenk@polynet.lviv.ua (A.Z.)

* Correspondence: mariyarkozak@gmail.com

**Abstract:** Vaccination is one of the main methods for the specific prevention of infectious diseases. The disadvantage of vaccination is the use of pathogens (live or attenuated viruses and bacteria) that can lead to the development of a disease. Recombinant technologies are capable of producing specific DNA or protein molecules that possess antigenic properties and do not cause disease. However, individual antigen molecules are low-immunogenic, and therefore, require conjugation with a compound possessing stronger immunogenic properties. In this study, we examined the immunogenic properties of the new anionic copolymer consisting of glycidyl methacrylate, butyl acrylate, triethylene glycol dimethacrylate, and acrylic acid, in mice. The experimental polymer induced a stronger immunogenic response than aluminum hydroxide. The histological studies have established that immunization both with aluminum hydroxide and the polymer studied does not cause damage to the liver, kidneys, or the spleen. No negative side effects were observed. It has been concluded that the new synthetic anionic polyelectrolyte hydrogel (PHG) has a potential as an adjuvant for vaccine development.

**Keywords:** adjuvant; anionic polyelectrolyte hydrogel; aluminum hydroxide; vaccine

## 1. Introduction

Vaccination is the best way to protect organisms from infectious diseases. Vaccines improved the quality of life, work productivity, and social equity of a population by averting untold numbers of disabling post-infection disease sequelae, most notably blindness, deafness, and neurological disorders [1–3]. However, no vaccine is 100% safe. Vaccines can have various side effects such as allergy, muscle pain, headache, swelling, shivering, and mild fever. In rare cases, live attenuated viral or bacterial vaccines can cause infection or morbidity.

Safe individual antigen molecules such as specific DNA sequences or proteins have low immunogenicity and, therefore, require conjugation with a compound possessing stronger immunogenic properties. An adjuvant, from the original Latin word meaning "helping" or "aiding", is an agent that amplifies the effect of a vaccine. Le Moignic and Pinoy described the first adjuvant—an emulsion of mineral oil and lanolin—in 1916 [4].

The adjuvant properties of aluminum compounds were first discovered by Glenny in 1926 [5] Since this discovery, aluminum phosphate and aluminum hydroxide have been frequently utilized in vaccine development due to their safety and efficacy in human vaccines. However, limitations of aluminum adjuvants include local reactions, production of IgE antibodies, ineffectiveness for some antigens, and an inability to elicit cell-mediated immune responses especially cytotoxic T-cell responses [6]. In 1935, Freund created a water-in-oil emulsion containing killed mycobacteria—complete Freund's adjuvant (CFA) [7]. CFA is among the most effective adjuvants, but it is also toxic and, therefore, not suitable for

human vaccines. CFA induces local reactions and granulomas, inflammation, and fever [8]. On the other hand, incomplete Freund's adjuvant (IFA) was successfully used in several vaccines licensed in the UK [9]. However, the use of IFA in humans was reported to induce strong side effects such as sterile abscesses [10]. MF59 oil-in-water emulsion of squalene oil and surfactants Span 85 and Tween 80 in citrate buffer is a safe and effective vaccine adjuvant which was originally approved to be included in a licensed influenza vaccine for elderly people in Europe in 1997 [11]. The MF59 adjuvanted influenza vaccine (Fluad™) is now licensed in more than 20 countries, and more than 85 million doses have been administered [12]. Fluad is an inactivated influenza vaccine indicated for active immunization against influenza caused by influenza virus subtypes A and type B contained in the vaccine [13]. Its common side effects include: injection site reactions (redness, tenderness, swelling, skin discoloration, localized warm skin, and a hard lump), fever, headache, fatigue, general feeling of being unwell (malaise), muscle pain, rash, sweating, joint pain, chills, diarrhea, nausea, vomiting, and changes in appetite [14].

The selection or synthesis of an appropriate adjuvant is the first step in developing a safe vaccine. Among other adjuvants that have been successfully tested in preclinical studies, an ionic polyelectrolytes, the ternary copolymer of 1,4-ethylenepiperazine, 1,4-ethylenepiperazine-*N*-oxide, and (*N*-carboxymethylene)-1,4-ethylenepiperazinium bromide ("Polyoxidonium"$^{\circledR}$) is permitted for human administration [1,15].

The aim of this work is to find an adjuvant which can be connected only with antigen molecules for the development of a vaccine with increased safety profile. Subunit vaccines serve as low risk alternatives to conventional vaccines because they have no potential for infection and each component can be tested for its toxicity [16].

In this study, we assessed a novel anionic polyelectrolyte hydrogel (PHG), a copolymer of glycidyl methacrylate, butyl acrylate, triethylene glycol dimethacrylate, and acrylic acid, as a potential adjuvant for vaccine development and analyzed effects of immunization with PHG as an adjuvant in mice. Its properties have been compared to aluminum hydroxide, a component of many commercial vaccines.

## 2. Materials and Methods

### 2.1. Materials

Monomers used to synthesize anionic PHG particles, acrylic acid (AA) (Aldrich, Milwaukee, WI, USA), glycidyl methacrylate (GMA) (Aldrich, USA), and butyl acrylate (BA) (Merck, Darmstadt, Germany) were purified by distillation in vacuo. Triethylene glycol dimethacrylate (TEGDMA), the initiator azobisisobutyronitrile (AIBN), solvents heptane, ethanol and hexane (all from Merck, Darmstadt, Germany), ethyl acetate, acetone, and dioxane (all from Aldrich, Milwaukee, WI, USA) were used without additional purification.

### 2.2. Synthesis of Hydrogel Particles

In a typical experiment, AIBN (0.25 g, 1.52 mmol) was dissolved in a mixture of monomers: GMA (3.14 mL, 22.1 mmol), BA (4.51 mL, 35.2 mmol), AA(12.0 mL, 166.6 mmol), and TEGDMA (0.58 mL, 2.0 mmol) (total (monomers) = 4 mol/L), then the mixture was purged with argon. An amount of 30.3 mL heptane was added to the mixture, and the solution was again purged with argon. Glass round-bottomed dilatometers (volume 20 mL) with a graduated precision bore capillary (SIMAX, Sázava, Czech Republic) were used for polymerization. The solution of monomers in heptane was loaded into dilatometers, and the polymerization occurred at 70 °C for 6 h in an argon atmosphere. The monomer conversion measured using dilatometric and gravimetric techniques [17] was 85%. The residual monomers used for preparing a 3D cured gel as well as uncured polymer molecules were removed using ethanol, acetone, toluene, and water in a Soxhlet extractor [18]. After washing, the hydrogel particles were dried for 8 h at 60 °C under vacuum till constant weight.

### 2.3. Characterization of Hydrogel Particles

The functional composition of the polymer hydrogel suspended in deuterated dimethyl sulfoxide was confirmed by proton nuclear magnetic resonance ($^1$H NMR) spectroscopy. $^1$H NMR spectra were recorded on a Bruker Avance DPX 300 (Bruker, Billerica, MA, USA) spectrometer at 300.13 MHz.

The carboxyl groups content was determined by acid-base back titration and the content of epoxy groups—by reverse titration of the residues of hydrochloric acid with 0.1 N NaOH [18]. For the measurement of carboxyl group content in the gels, an excess of 0.1 N aqueous solution of NaOH (5 mL) was added to the particle dispersion (0.2 g) and stirred for 6 h at 60 °C, then insoluble gel particles were washed with water and separated using a centrifuge Hettich Mikro 22R (Andreas Hettich GmbH & Co. KG, Tuttlingen, Germany) to avoid sorption of the reagents by the gel particles, and the total amount of the separated liquid phase containing residual NaOH was titrated with 0.1 N solution of HCl.

The content of carboxyl groups (COOH) in the polyelectrolyte hydrogel particles (PHG) was calculated using the Equation (1):

$$[COOH]\% = (m_{NaOH} \cdot M_{COOH} \cdot 100)/(M_{NaOH} \cdot m_{PHG}) \tag{1}$$

where $m_{NaOH}$ (g) is the weight of NaOH that reacted with carboxyl groups; $M_{COOH}$ and $M_{NaOH}$—the molar weights of carboxyl group and NaOH molecule, respectively; $m_{PHG}$ (g)—weight of analyzable hydrogel particles.

The content of acrylic acid units (AA-units) in the PHG:

$$[AA\text{-units}]\% = ([COOH]\% \, M_{AA})/M_{COOH} \tag{2}$$

where $M_{AA}$ and $M_{COOH}$ are the molar weights of acrylic acid and carboxyl group, respectively.

For the measurement of epoxy group (EP—groups) content in the PHG an excess of the solution of 0.1 N HCl in acetone (10 mL) was added to the particle dispersion (0.2 g) and stirred for 6 h, then insoluble gel particles were washed with water and separated by centrifugation, and the total amount of the separated liquid phase containing residual HCl was titrated with 0.1 N aqueous solution of NaOH.

The content of epoxy groups in the PHG particles was calculated using the Equation (3):

$$[EP\text{-groups}]\% = (V_{NaOH} \cdot 0.0043 \cdot 100)/m_{PHG} \tag{3}$$

where $V_{NaOH}$ (mL)—volume of NaOH solution consumed to titrate the analyzable sample; 0.0043—the weight (g) of epoxy groups corresponding to 1 mL of exactly 0.1 N NaOH; $m_{PHG}$ (g)—weight of hydrogel particles.

The content of glycidyl methacrylate units (GMA-units) in the PHG:

$$[GMA\text{-units}]\% = ([EP\text{-groups}]\% \cdot M_{GMA})/M_{EP} \tag{4}$$

where $M_{GMA}$ and $M_{EP}$—molar weights of GMA and the epoxy fragment, respectively.

The size of PHG particles separated from aqueous dispersion was measured by a transmission electron microscope JEM-200A (JEOL, Tokyo, Japan) at an accelerating voltage of 200 kV. The samples of the organic and aqueous dispersion were prepared by dispersing particles in bidistilled water or toluene; the concentration of microgel particles was 10 mg/mL. Samples were prepared [19] by spraying the tested solution on a substrate with an ultrasonic dispersant UZDN-1A (UkrRosPribor Ltd., Sumy, Ukraine), which created a uniform coating of the substrate. A thin amorphous carbon film deposited on a copper grid was used as the substrate [20].

The hydrodynamic diameter of the hydrogel particles and their zeta-potential were measured by dynamic light scattering (DLS) using the Zetasizer Nano (Malvern Instruments Ltd., Worcestershire, UK) device at 25 °C. The particle concentration in the samples was 0.4 mg/mL. Five measurements were

made for every sample (each measurement consisted of five cycles, the range between measurements was 5 min). The stability of aqueous dispersions of particles was confirmed by measuring their hydrodynamic radius by the DLS method for 5 h.

### 2.4. Immunization

Mice immunization was performed using white laboratory BALB/c mice, both male and female. Animals of five months of age (150–160 days) were divided into three groups of five mice each. Mice were injected subcutaneously (site for injection is over the shoulders). Animals of the first experimental group were injected with a mixture (100 µL) of the PHG solution (40 mg/mL) and bovine serum albumin (BSA) (Sigma-Aldrich, Darmstadt, Germany) at a concentration of 100 mg/mL, in a 1:1 volume ratio; mice of the second experimental group were injected with a solution (100 µL) of 80 mg/mL of aluminum hydroxide Al(OH)$_3$ (Sphera Sim, Lviv, Ukraine) with BSA at a concentration of 100 mg/mL, in a 1:1 volume ratio. Aluminum hydroxide particle size was 1200 ± 270 nm. The particle size was measured by dynamic light scattering method on a DynaPro NanoStar analyzer (Wyatt Technology, Santa Barbara, CA, USA). A control group of mice were injected with 100 µL of a 0.9% solution of sodium chloride. Immunization was performed on days 1, 14, and 28. The solutions were sterilized by autoclaving. One week after the final injection (day 35), animals were anesthetized in an induction chamber for 5 s using chloroform (Sphere Sim, Ukraine), decapitated by cervical dislocation and their blood was taken for analysis. Blood was collected in 1.5 mL Eppendorf tube containing 7.5 uL of heparin (1000 u/mL, Sigma-Aldrich #H3393).

All animals were maintained in a specific pathogen-free animal facility with water and commercial food (Rokki 1, Rivne, Ukraine) provided *ad libitum*. The protocol for animal experiments was approved by the Ethical Committee of the Institute of Animal Biology of NAAS of Ukraine and the experiments were carried out in accordance with the European Convention for the Protection of Vertebrate Animals (Strasbourg, French, 1986).

### 2.5. Antibody Purification and Detection

Immunoglobulins were isolated from blood serum of mice by precipitating three times with a saturated solution of ammonium sulfate (Sphere Sim, Ukraine)**.** Immunoglobulins were then dissolved in phosphate buffered saline (PBS) for use in an ELISA method. ELISA method: 100 µL of 1% BSA solution was adsorbed onto a 96-wellplate (PAA, Pasching, Austria), incubated for 24 h at 4 °C; the wells were then washed three times with buffer A (0.2% BSA in PBS) and the isolated immunoglobulins in PBS solution were added to the wells and incubated for 2 h at 37 °C, then washed three times with buffer A; conjugated anti-mouse antibodies (Sigma, Darmstadt, Germany) were added in a 1:5000 dilution, incubated for 1 h at 37 °C; the wells were then washed three times with PBS-Tween-20, and the substrate for alkaline phosphatase, *p*-nitrophenylphosphate in diethanolamine (Filisit-Diagnostics, Dnipro, Ukraine) was added; after incubating at room temperature for 3 min, absorbance at 405 nm was measured on an ELISA plate reader (STAT FAX Awareness Technology Inc., Palm, FL, USA). Dot blot analysis: 2 µL of a 10% solution of BSA was dripped onto a nitrocellulose membrane and blocked by skimmed milk powder for 10 min; after completely drying the drop, mice immunoglobulins were applied and incubated for 30 min at room temperature; washed with a Tris-buffered saline (TBS) + milk powder + Tween-20; incubated with anti-mouse antibodies (A3562 Sigma, Germany) for 30 min at room temperature; washed with a buffer (TBS + Tween-20); signal detection was carried out with a substrate for alkaline phosphatase CDPstar (Amersham, Foster, CA, USA) on X-ray filmICL hyperFilm (Amersham, Foster, CA, USA).

### 2.6. Histopathological Characterization

Organs (liver, spleen and kidneys) are first inspected with the naked eye (macroscopy) and described (size and number of tissue fragments, visual abnormalities, color). For microhistological studies, 0.2–0.3 cm sections of the liver, spleen and kidneys were fixed in 10% formalin solution at

room temperature. After that, they were rinsed in running water for 30–60 min, slowly dehydrated with grades of ethanol (70, 80, 90, 95, and 100%). Dehydration was then followed by clearing the samples in two changes of xylene, and the paraffin blocks were formed using a standard method [21]. Sections with a thickness of 7 μm were made on a microtome (Microm HM 340 E) and stained with hematoxylin-eosin [21]. The mounted specimens were observed and were scored under light microscopy, and the photomicrographs of them were obtained.

*2.7. Particle Toxicity Test*

Mice immunization was performed using male white laboratory BALB/c. Animals of five months of age (150–160 days) were divided into three groups of five mice each. Mice were injected subcutaneously (site for injection is over the shoulders). Animals of the first experimental group were injected with 50 μL of the 40 mg/mL PHG in PBS, pH 7.2; mice of the second experimental group were injected with with 50 μL of the 400 mg/mL PHG in PBS. The control group of mice were injected with 50 μL of PBS. Immunization was performed on days 1, 14, and 28. The solutions were sterilized by autoclaving. One week after the final injection (day 35), animals were weighed and anesthetized in an induction chamber for 5 s using chloroform (Sphere Sim, Lviv, Ukraine), decapitated by cervical dislocation. The liver, kidneys, and spleen were removed and weighed. Organ-to-body weight ratios were calculated.

*2.8. Statistical Analysis*

Statistical calculations of results (M ± m) were performed using Microsoft Excel 2007. The probability of differences was determined by the Student's *t*-test with $p < 0.05$ accepted as statistically significant.

## 3. Results

*3.1. The Synthesis, Structural, and Colloidal-Chemical Properties of Polyelectrolyte Based Cured Hydrogel Particles*

Polymer hydrogel 3D cured particles were prepared using the precipitation copolymerization of Glycidyl methacrylate (GMA), butyl acrylate (BA), Acrylic acid (AA), and Triethylene glycol dimethacrylate (TEGDMA) as (co)monomer and a curing agent simultaneously. The scheme of the synthesis of 3D cured gel particles presents the formation first of a linear polymer and subsequent formation of cured gel particles (Figure 1).

The characteristics of synthesized cured hydrogels are presented in the Table 1.

**Table 1.** Compositions and colloidal-chemical characteristics of synthesized cured hydrogels particles (PHG).

| Composition of the Monomer Mixture, Mole Fraction, % | | | | Composition of the Polymer Hydrogel, Mole Fraction, % | | | | Average Particle Size, nm (from TEM) | Hydrodynamic. Diameter of Particles (Dispersion in $H_2O$), nm (from DLS) | Zeta-Potential, mV |
|---|---|---|---|---|---|---|---|---|---|---|
| GMA | BA | TEGDMA | AA | GMA | BA | TEGDMA | AA | | | |
| $k'$ | $l'$ | $m'$ | $n'$ | $k$ | $l$ | $m$ | $n$ | | | |
| 10 | 15 | 1 | 74 | 14 | 12 | 4 | 70 | 400 ± 130 | 600 ± 250 | −49 |

It is obvious (Figure 2) the evolution of the colloidal polymer system at different stages of precipitation polymerization from transparent solution of initial monomer mixture to highly stable dispersion of cured gel particles at 85–90% monomer conversion.

As is evident from the scheme (Figure 1), the definite amount of carboxyl groups in hydrogel particles provide the particle hydrophily and stability in water dispersions, particularly at pH value near 7.2–7.4, as well as the definite amount of reactive epoxide groups capable of covalent binding proteins. The functional composition of synthesized particles was confirmed by functional analysis (Table 1), as well as using NMR spectroscopy (Figure 3). The polymeric skeletal $CH_2$ at 2.1–1.7 ppm,

skeletal CH at 3.4–3.2 ppm; the fragments—CH$_3$ repeated in TEGDMA and GMA units at 1.6–1.2 ppm; signal of protons of fragment C(O)-O-CH$_2$—repeated in TEGDMA, BA, and GMA units at 3.9–3.8 ppm; the fragment -CH-CH$_2$-O- (GMA units) at 2.7 ppm; the fragment -CH$_2$-CH$_3$ (BA units) at 1.6 and 0.96 ppm, respectively; the fragment -COOH (AA units) at 10.65 ppm; and the fragment -CH$_2$-CH$_2$-O- (TEGDMA units) at 3.7 ppm are present in the $^1$H-NMR spectrum (Figure 3). The low signal at 5.5 ppm can be attributed to protons of free unreacted methacrylate groups of TEGDMA.

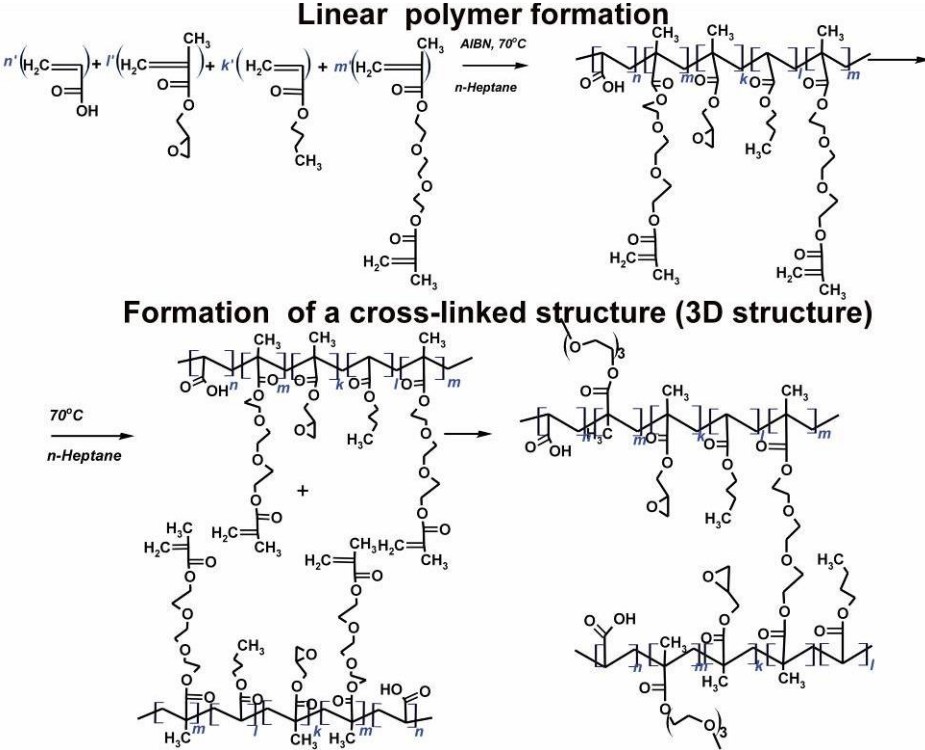

**Figure 1.** General scheme of the synthesis of poly(GMA-co-BA-co-TEGDMA-co-AA) 3D cured hydrogels.

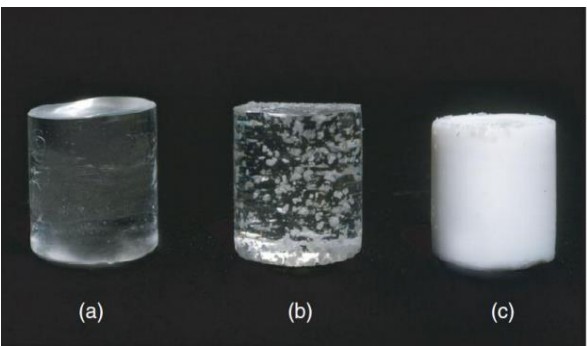

**Figure 2.** The images of gel dispersion formation at precipitation polymerization in hexane: initial solution of the monomers (**a**), at monomer conversion 25–30% (**b**), and at monomer conversion 85–90% (**c**).

One can see from TEM images (Figure 4) and the DLS study (Figure 5) that hydrogel particles are of a spherical shape, predominantly sized in the range 500–900 nm. An average hydrodynamic diameter of the particles is ~600 nm. As expected, the hydrodynamic diameters from the DLS measurements were substantially larger comparing the size determined using TEM image. It is also evident (Figure 5) that ionized carboxyl-containing particles are charged negatively in phosphate buffered saline and PBS

and provide high enough sedimentation and aggregation stabilities of the water dispersions during 5 h at least.

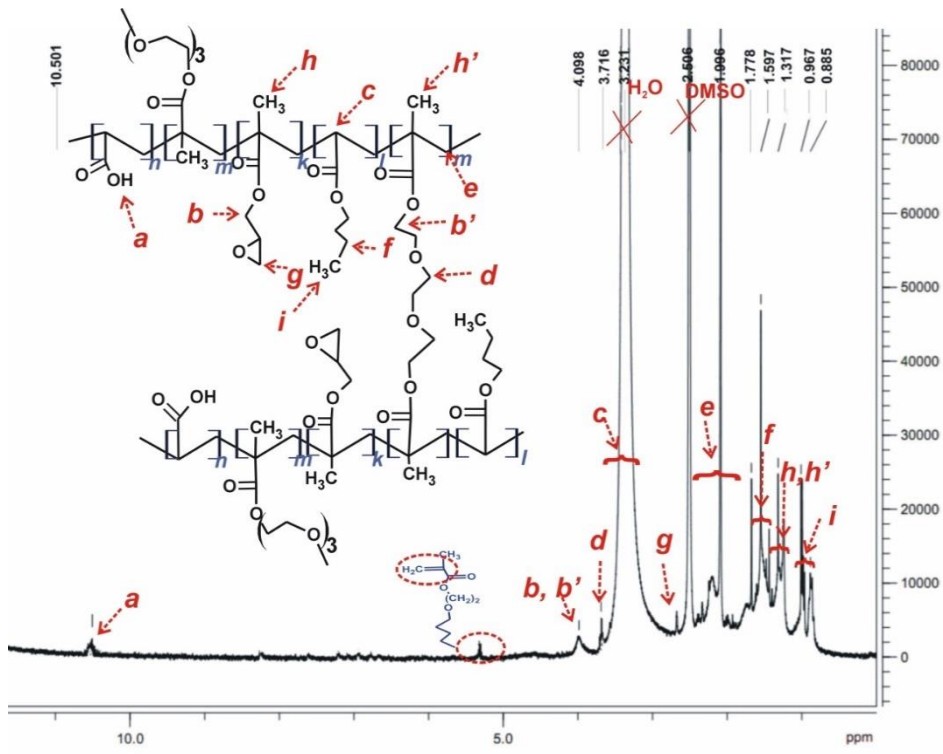

**Figure 3.** $^1$H NMR spectra of poly(GMA-co-BA-co-TEGDMA-co-AA) hydrogel particles.

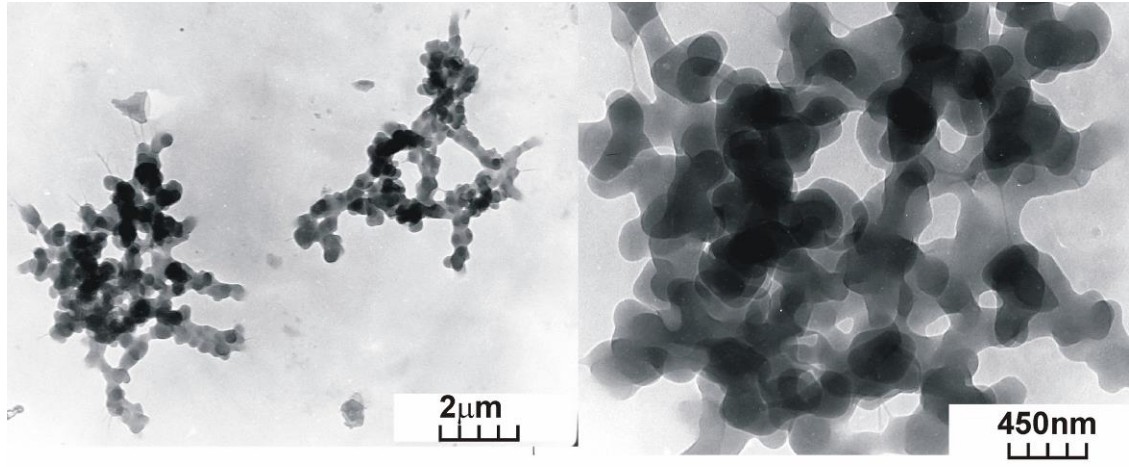

**Figure 4.** TEM images of PHG particles at different magnifications: ×10,000 (**left**) and ×30,000 (**right**).

### 3.2. Histology Observations

We studied the structure of the spleen, liver, and kidneys. Macroscopic observation indicated that the spleen of the control and experimental groups of mice was of elastic consistency and sickle shaped. During the microscopic examination of tissues from the control group, capsules with trabeculae, white and red pulp, trabecular veins and arteries, and lymph nodes (Figure 5) were clearly visible.

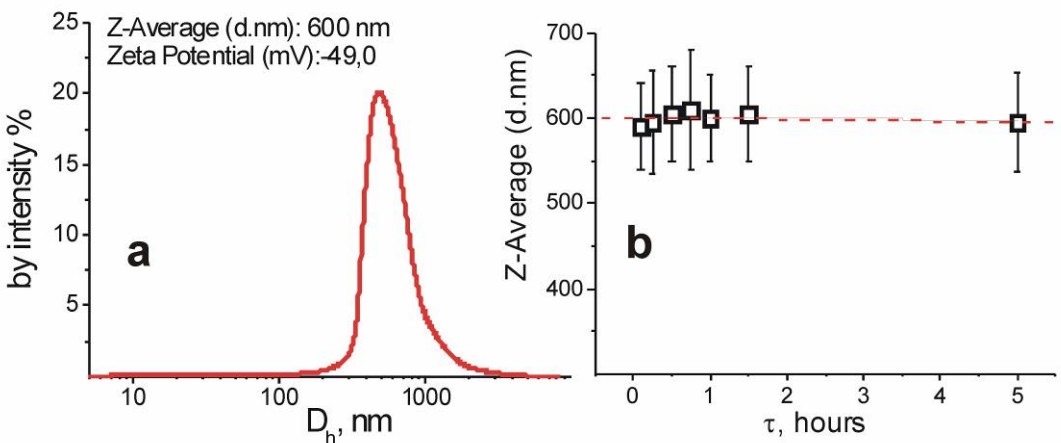

**Figure 5.** DLS study of PHG particle size Z-potential (**a**) and their aggregation stability in water dispersion (**b**).

### 3.3. Adjuvant Properties of PHG

Adjuvant properties of PHG were compared to aluminum hydroxide, a component of many licensed vaccines. The experimental polymer outperformed aluminum hydroxide. We observed a 2-fold increase in specific antibodies against BSA after immunization with PHG compared to aluminum hydroxide (Figure 6).

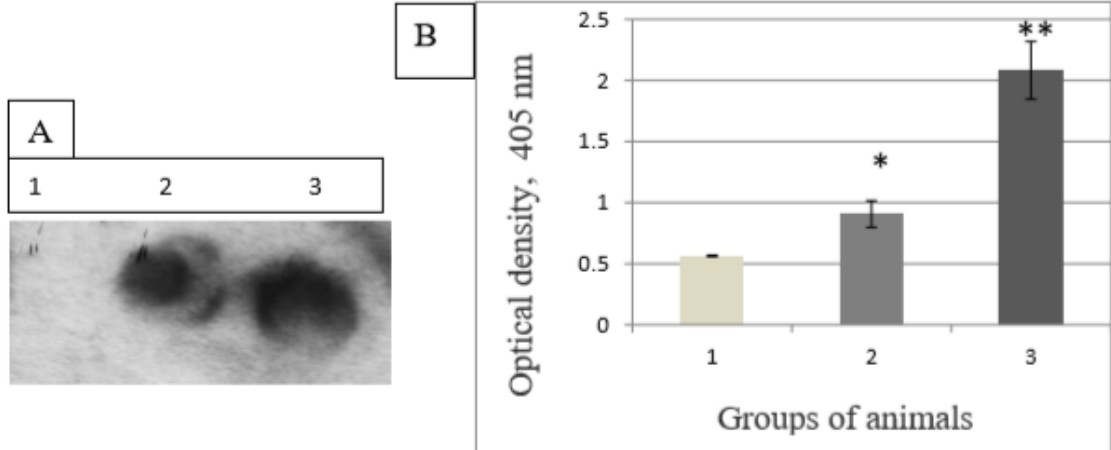

**Figure 6.** Dot blot (**A**) and ELISA (**B**) analysis of anti BSA antibodies level in the serum of immunized mice: 1—control (sodium chloride), 2—1st experimental group (aluminum hydroxide + BSA), 3—2nd experimental (polymer hydrogel + BSA). Statistically significant results at * $p \leq 0.05$ and ** $p \leq 0.01$.

Trabeculae thinning (Figure 7) was observed in mice after immunization with aluminum hydroxide. Trabecular veins and arteries were poorly differentiated. Increases in the number of red blood cells, platelets, monocytes, and lymphocytes in the red spleen pulp were observed in mice immunized with aluminum hydroxide. White spleen pulp was poorly developed in mice immunized with PHG.

The macroscopic liver structure of the control and both experimental groups was of solid consistency, and reddish-brown in color. It was observed microscopically that the structure of the organs both of the control and experimental groups were unchanged.

The cytoplasm of hepatocytes was homogeneous in color, and nuclei were clearly visible. Chromatin was localized predominantly on the periphery of the nucleolus. Erythrocyte remains were found in the veins. No structural or pathological changes were observed in the experimental groups.

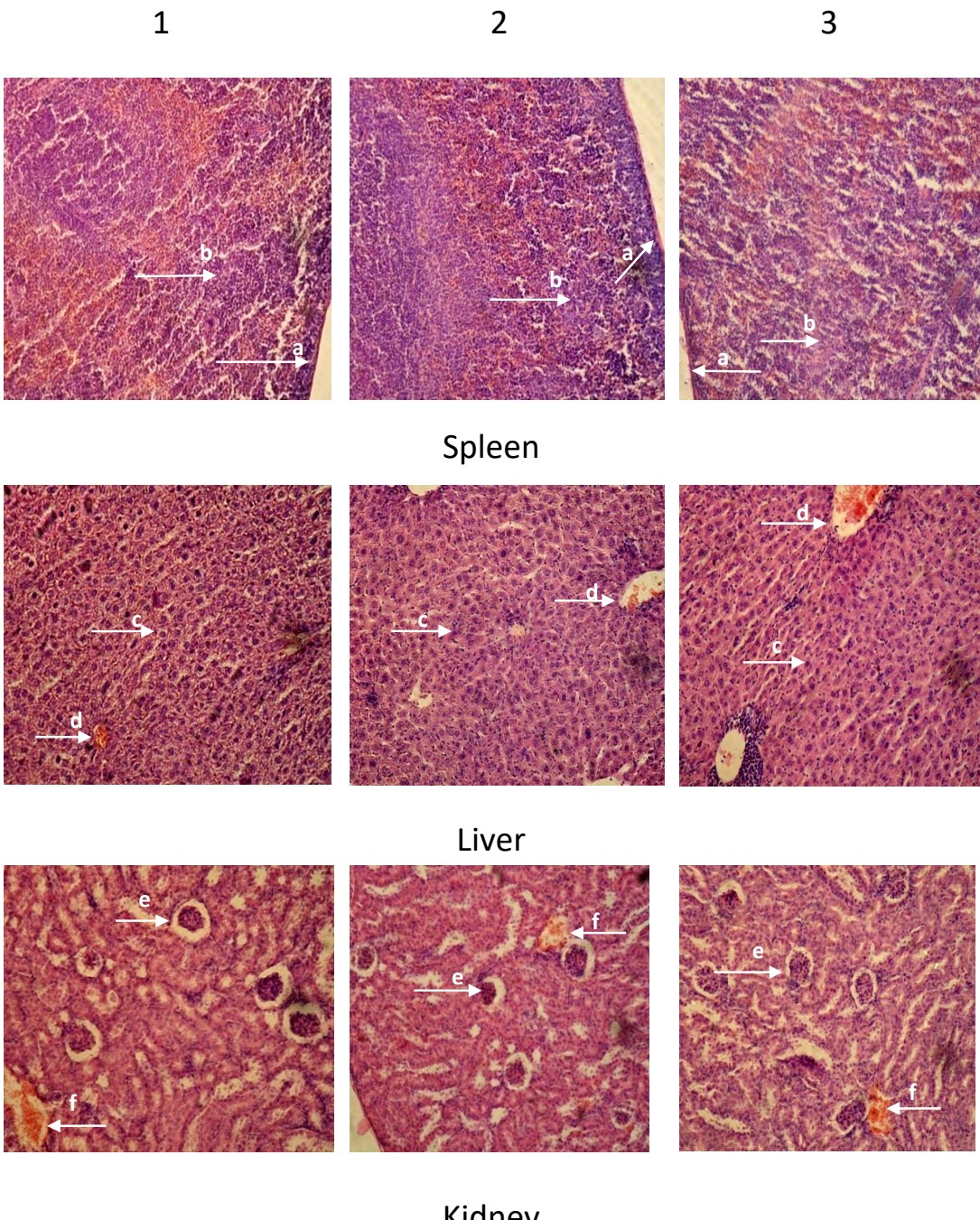

**Figure 7.** Microstructure of the organs of white mice (Hematoxylin—eosin, ×100). (**1**)—control; (**2**)—polymer; (**3**)—aluminum hydroxide; a—capsule; b—lymphatic follicle; c—hepatocytes; d—a vein filled with erythrocytes; e—vascular glomerulus; f—vessels with red blood cells.

Macroscopic examination showed solid consistency and reddish-brown color in the kidneys of mice in both the control and experimental groups. The renal cortex and renal medulla were clearly distinguished on histological sections (Figure 7).

The structure of the kidney tissue was preserved. There were no hemorrhages, tubular epithelium did not change, lumen was clearly visible, and the vascular glomerulus was of normal size (Figure 5). Vessels were moderately blood-filled, and the epithelium was without visible changes.

### 3.4. Body Weight and Coefficient of Organs

Organ weight changes have long been accepted as a sensitive indicator of chemically induced changes to organs. There were no significant differences in food and water intake in the PHG-treated animals compared to control groups. The organ to body weight ratios of the liver, kidney, and spleen of animals treated with lower (40 mg/mL) dose did not show any significant differences from the control group (Figure 8), whereas in animals treated with 20 mg of PHG per mouse (400 mg/mL), ratio of the kidney to body weight increased significantly. There were no marked changes in liver and spleen to body weight ratios.

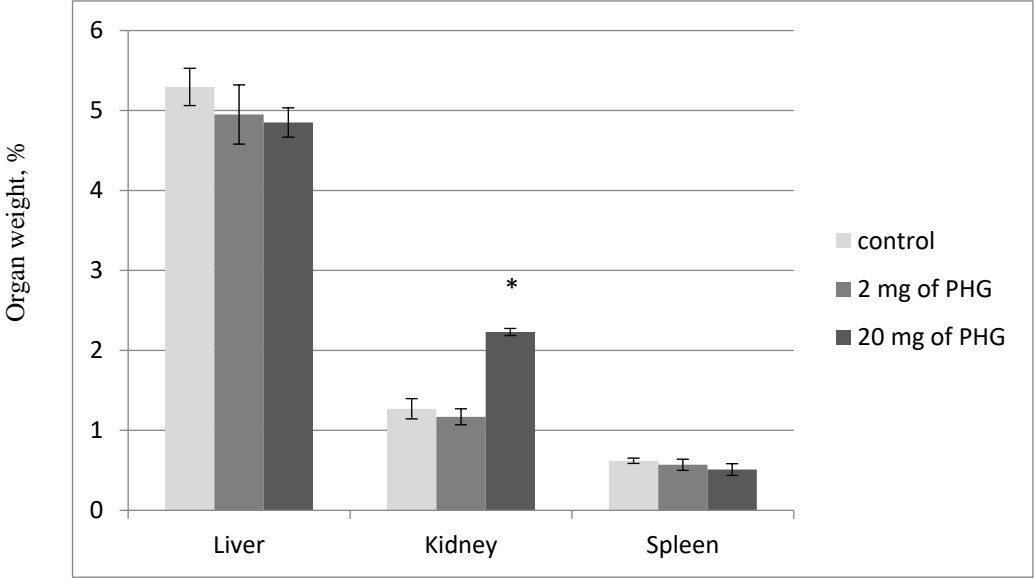

**Figure 8.** Organs to bodyweight ratio (liver, kidney, and spleen). * Statistically significant results at $p \leq 0.05$.

## 4. Discussion

Polymeric compounds of different composition were studied for their adjuvant properties. Characteristics of adjuvants were established for nanoparticles (NPs) of poly(methyl vinyl ether-co-maleic anhydride) copolymer (PVM-MA), Gantrez™ AN119 polymers coated with vitamin B12 [22], poly(methyl vinyl ether-co-maleic anhydride) [23], poly(methyl methacrylate) [24] in HIV vaccines, cytosine-phosphate-guanosine oligodeoxynucleotide [25], chitosan [26], and many others [27]. Adjuvant properties of functional polymer coated polystyrene nanoparticles and polymers based on acrylic acid were described by Kit et al. and Kozak et al. [28,29]. This study demonstrated potent adjuvant properties of PHG. PHG adjuvant effect was greater than that of aluminum hydroxide, the adjuvant used in many commercial vaccines. Increased titers of anti-BSA antibodies when PHG was used as an adjuvant may be explained by: (1) its size of 0.640 μm; (2) attachment of antigen molecules, which are considered to be slowly eliminated from the site of injection similar to aluminum compounds[1]. Particles in the size range of 20–200 nm are efficiently taken up by dendritic cells via endocytosis or pinocytosis and facilitate the induction of cellular immune responses, whereas microparticles of 0.5–5 μm are taken up via phagocytosis or macropinocytosis, mainly generating humoral responses [30]. A study investigating the immunogenicity of differently sized particles (200, 500, and 1 μm) encapsulating BSA showed that 1 μm-sized poly(lactic-co-glycolic acid particles) were capable of inducing stronger IgG responses in vivo than 200 and 500 nm nanoparticles following immunization via intranasal, oral, and subcutaneous routes in mice [31]. We hypothesize that a maximum immune response depends on the adjuvant particle size. The particle size of aluminum hydroxide adjuvant is 1200 ± 270 nm. Our previous experiments showed 33.3% less effective

immunization with PHG (2.12 μm) which was the same in terms of the elemental chemical composition as PHG (0.640 μm) but differed in quantitative ratios of monomers and the particle size [32].

Subcutaneous injections of the investigated PHG polymer did not lead to visible damage of animals throughout the observation period. There was no swelling or redness at the site of immunization as observed with complete Freund's adjuvant [28], the animal hair remained dense and shiny, and the animal weights were stable. Lethal cases among tested animals were absent.

Previously, it was shown that the aluminum hydroxide gel adjuvanted rabies vaccine had a significant effect on kidneys in the area of the cortex, arterial wall, and highly distorted glomeruli which contained many pyknotic or karyolitic nuclei [33]. The compelling effect on the liver in the portal area showed an increase in Kupffer cells with some normal architecture in hepatocytes and central vein but few hepatocytes contain pyknotic nuclei. The histopathological examination indicated that significant inflammatory reactions accompanied monophosphoryl lipid A adjuvant vaccine compared to aluminum hydroxide gel adjuvant vaccine [33].

Based on the histological studies performed, it was found that the structures of the kidneys and liver were without pathological changes in animals after immunization with both aluminum hydroxide and PHG. Organ weights are an applicable screening tool to identify treatment-related effects. Organ-to-body weight ratios are commonly calculated and are considered more useful. We found an increased kidney relative weight only in mice after their immunization with 10-fold higher dose (20 mg of PHG per mouse) than necessary for a specific antibody production (2 mg per mouse). This increase is within normal kidney-to-body ratio for 5-10-month-old Balb/c mice [34].

In conclusion, we synthesized copolymer hydrogel with average particle size of 640 nm, containing 14% of GMA, 12% of BA, 4% of TGM, and 70% of AA. PHG was found to increase anti-BSA antibody titer by 2-fold in mice substantially outperforming aluminum hydroxide adjuvant. The histological observations showed that the PHG immunization did not cause damage to the liver, kidneys, or spleen of mice. It can be concluded that PHG is effective as an adjuvant and has a potential for vaccine development.

**Author Contributions:** M.K., V.V. and A.Z. conceived and designed the experiments; A.Z. and N.M. performed the chemical experiments and analyzed the data; M.K. performed the biological experiments and analyzed the data; A.Z., N.M. and M.K. wrote the paper. All authors have read and agreed to the published version of the manuscript.

**Funding:** This research received no external funding.

**Acknowledgments:** The authors would like to acknowledge the United States Department of Defense, Defense Threat Reduction Agency (DTRA), and Ukraine Biological Threat Reduction Program (BTRP) for their assistance and financial support in publication of this paper. The authors thank S. Mykhalovsky for help with manuscript preparation. While DTRA/BTRP did not support the research described in this publication, the program supported the manuscript development and publication. The contents of this publication are the responsibility of the authors and do not necessarily reflect the views of DTRA or the United States Government. The authors wish also to thank Yu. Martin and A. Oliynyk (Institute of Animal Biology, National Agrarian Academy of Sciences (NAAS) of Ukraine) for their help in histology studies and M. Moskvin (Lviv Polytechnic National University) for his assistance in PHG synthesis.

**Conflicts of Interest:** The authors declare no conflict of interest.

## Abbreviations

| | |
|---|---|
| AA | acrylic acid |
| AIBN | initiator azobisisobutyronitrile |
| BA | butyl acrylate |
| BSA | bovine serum albumin |
| CFA | complete Freund's adjuvant |
| DLS | dynamic light scattering |
| GMA | glycidyl methacrylate |
| 1H NMR | proton nuclear magnetic resonance |

## Abbreviations

| | |
|---|---|
| IFA | incomplete Freund's adjuvant |
| NPs | nanoparticles |
| PBS | phosphate buffered saline |
| PHG | anionic polyelectrolyte hydrogel |
| TBS | Tris-buffered saline |
| TEGDMA | Triethylene glycol dimethacrylate |

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
