# Peer review of "Anionic Polyelectrolyte Hydrogel as an Adjuvant for Vaccine Development"

_scipharm, doi:10.3390/scipharm88040056_

Round 1

Reviewer 1 Report

This manuscript presents the evaluation of a novel anionic polyelectrolyte hydrogel (PHG), a copolymer of glycidyl methacrylate, butyl acrylate, triethylene glycol dimethacrylate and acrylic acid, as a potential adjuvant for vaccine development and analyzed effects of immunization with PHG as an adjuvant on mice. Its properties have been compared to aluminum hydroxide, a component of many commercial vaccines.

The experiments in this study have serious defects, additional experiments are needed:

  • the equations are written in the text and are difficult to follow
  • the methods of obtaining and characterisation of the particles are poorly presented
  • histological and immunization tests are not presented in detail
  • particle toxicity tests should be introduced

Author Response

Dear Reviewer,

We would like to thank You for the detailed analysis of the presented results.

Please, find our answers on your remarks in the attachment.

On behalf of all co-authors,

Yours sincerely,

Mariya Kozak

Reviewer 2 Report

The work describes the synthesis of anionic polyelectrolyte hydrogel and its potential application as a vaccine adjuvant. The authors claim the improvement of the immune response compared to the adjuvant used commercially. Despite the novelty of the hydrogel for its use as an adjuvant, the work lacks quality both in the presentation and in the quality of the figures. Below are some of my comments and suggestions for improvement ahead of publication.
1. The abstract should not include some introductory remarks that should be in the introductory section and the abstract should be revised and rewritten.
2. The introduction should also be revised as some paragraphs should be merged. Also, the last three paragraphs should be ordered correctly.
3. Authors should also consider discussing all the results of the work and avoid putting some of the results in the materials and methods section. All results should be presented in the results and / or included in the results and discussion section.
4. The quality of the TEM figure is very poor and should be replaced with a better figure.

5.Enhancement of immune response from micro sized adjuvants was also previously reported else where . how is the current work compared with other reports?

6. I have noticed many abbreviations and I recommend putting a list of abbreviations at the end of the work.

7. check reference 8

Author Response

(The authors gave the same response as above.)

Round 2

Reviewer 1 Report

The manuscript has been improved according to my suggestions and that is why I agree with its publication